# The Interplay between MicroRNAs and the Components of the Tumor Microenvironment in B-Cell Malignancies

**DOI:** 10.3390/ijms21093387

**Published:** 2020-05-11

**Authors:** Sherien M. El-Daly, Recep Bayraktar, Simone Anfossi, George A. Calin

**Affiliations:** 1Department of Experimental Therapeutics, The University of Texas MD Anderson Cancer Center, Houston, TX 77030, USA; sm.el-daly@nrc.sci.eg (S.M.E.-D.); rbayraktar@mdanderson.org (R.B.); sanfossi@mdanderson.org (S.A.); 2Department of Medical Biochemistry, Medical Research Division, National Research Centre, Cairo 12622, Egypt; 3Center for RNA Interference and Non-Coding RNAs, The University of Texas MD Anderson Cancer Center, Houston, TX 77030 USA; 4Department of Leukemia, The University of Texas MD Anderson Cancer Center, Houston, TX 77030, USA

**Keywords:** B-cell malignancies, tumor microenvironment, stroma, microRNAs, cell-to-cell communication, exosomal miRNAs, immune cells, endothelial cells, cancer-associated fibroblasts

## Abstract

An increased focus is being placed on the tumorigenesis and contexture of tumor microenvironment in hematopoietic and solid tumors. Despite recent clinical revolutions in adoptive T-cell transfer approaches and immune checkpoint blockade, tumor microenvironment is a major obstacle to tumor regression in B-cell malignancies. A transcriptional alteration of coding and non-coding RNAs, such as microRNAs (miRNAs), has been widely demonstrated in the tumor microenvironment of B-cell malignancies. MiRNAs have been associated with different clinical-biological forms of B-cell malignancies and involved in the regulation of B lymphocyte development, maturation, and function, including B-cell activation and malignant transformation. Additionally, tumor-secreted extracellular vesicles regulate recipient cell functions in the tumor microenvironment to facilitate metastasis and progression by delivering miRNA contents to neighboring cells. Herein, we focus on the interplay between miRNAs and tumor microenvironment components in the different B-cell malignancies and its impact on diagnosis, proliferation, and involvement in treatment resistance.

## 1. Introduction

The research on the pathogenesis of hematologic malignancies has been recently centered on the collaborative interaction between malignant cells and tumor microenvironment (TME). Such reciprocal interaction is recognized to play an essential role sustaining the different hallmarks of cancer from tumor proliferation, invasion, metastasis, and even participating in treatment resistance [1,2,3,4].

The TME is a highly heterogeneous environment in regards to its composition (cellular and non-cellular components) and the spatial arrangement of stromal cells [5]. The cellular components of TME consist of a large variety of stromal cells that include: follicular dendritic cells (FDCs); cancer-associated fibroblasts (CAFs); mesenchymal stem cells (MSCs); immune and inflammatory cells, such as tumor-associated macrophages (TAMs) or type 2 macrophages (M2); regulatory T-cells (Treg); dendritic cells (DCs); and tumor-infiltrating lymphocytes (TILs). On the other hand, the non-cellular components include structural matrix and soluble factors, such as cytokines, growth factors, small RNAs, and DNA [6,7]. The diversity in the cellular and non-cellular components in the TME varies according to the tumor genotype and/or phenotype [7]. The development and progression of some tumor types largely rely on the crosstalk between tumor cells and some of the TME components. Studies revealed that growth factors and different chemokines secreted by tumor cells could recruit stromal cells and educate them to generate a favorable microenvironment for tumor hosting and thriving. The interaction of “educated” stromal cells with tumor cells as well as among themselves plays a critical role in enhancing tumor proliferation, metastasis, and even development of drug resistance [8,9].

The development of novel drugs able to target the tumor-stroma interactions, prevent the connection of tumor cells to specific niches, or block the immune checkpoint regulatory proteins to promote tumor immune-surveillance, represents a potential strategy for effective cancer treatments. Encouraging results have already been shown in clinical trials [10,11,12,13]. Functions and characteristics of the TME may vary between different cancer types and even among patients with the same cancer type. Although TME of hematological malignancies is greatly different from that of solid tumors, the TME of lymphoma malignancies shares some characteristics from both solid and hematologic cancers [14].

## 2. Tumor Microenvironment of B-Cell Malignancy

Hematologic B-cell malignancies can occur at several stages during normal B-cell differentiation, including pre-germinal centers, germinal centers (GC), and post-GC B cells. Moreover, B-cell transformation involves multiple genetic events, which can activate oncogenes and disrupt the function of specific tumor suppressor genes after the alteration of immunoglobulin (Ig) gene rearrangements and somatic hypermutation of Ig variable region (V) genes [15,16]. In addition to these alterations, microenvironmental components that stimulate signals for B-cell growth and survival may also contribute to the development and progression of B-cell malignancies [17]. This is achieved by the number of signaling pathways that are involved in the initiation and development of B-cell lymphomagenesis.

Hematologic B-cell malignancies originate from uncontrolled growth of hematopoietic and lymphoid cells. These malignancies represent a clinically and biologically heterogeneous group of lymphoid neoplasms that include most Non-Hodgkin’s lymphomas (NHLs), chronic lymphocytic leukemia (CLL), and multiple myeloma (MM) and are characterized by the expression of one or more common B-cell antigens [18,19,20]. NHLs are generally divided based on the type of lymphocytes involved (B or T-lymphocytes), and further subdivided based on cell aggressiveness: aggressive (fast-growing) and indolent (slow-growing) lymphomas. The most common aggressive B-lymphomas include diffuse large B-cell lymphoma (DLBCL), Burkitt lymphoma (BL), mantle cell lymphoma (MCL), and lymphoblastic lymphoma (LL). On the other hand, common indolent B-cell lymphomas include follicular lymphoma (FL), lymphoplasmacytic lymphoma, marginal zone B-cell lymphomas (MZLs), small-cell lymphocytic lymphoma (SLL), and mucosa-associated lymphoid tissue (MALT) lymphoma [20]. CLL is considered to have the same entity as small lymphocytic lymphoma, with both being indolent and affecting the same cells. In CLL, the lymphoma cells are present in bone marrow and blood, but in SLL, lymphoma cells are found in the spleen and the lymph nodes [19].

The TME of hematologic malignancies is generally divided into three major models based on the dependency and interaction between tumor cells and their microenvironment components (Figure 1) [21]. The “re-education” model characterizes FL, CLL, MM, and MALT lymphomas, where the survival and proliferative drive for tumor cells are partially dependent on the external signals from the TME, such as antigens, cytokines, and cell contact with stromal cells. In this TME model, the cell arrangement and interaction between tumor cells and microenvironment largely resembles that of the normal lymphoid B cells within their respective niche [21,22].

The second pattern of interaction is featured in CHL, in which the tumor cells strongly rely on the supportive milieu of TME. Consequently, the survival and progression of this type of lymphoma show higher affinity for their microenvironment. This pattern is usually referred to as the “recruitment” model [21,22,23].

The third pattern (effacement) that is referred to as the loss of interconnection between malignant cells and their microenvironment, evolves mainly as a result of genetic mutations that provide malignant cells with microenvironment-independent growth and survival signals. This type can be mostly observed in BL, and to some extent in DLBCL, where myc proto-oncogene(c-myc) activation, resulting from chromosomal translocation, provides lymphoma cells with strong autonomous proliferation signals, with no dependency on microenvironment stimuli [21,22,24].

These types of classifications reveal that individual B-cell malignancies have distinctive microenvironment properties that should be considered in the therapeutic strategies. The interaction between lymphoma cells and microenvironment is mostly modulated by a genetic and epigenetic regulation. The major components of the epigenome include histone modifications, cytosine modifications, and regulatory functions of non-coding RNA (ncRNA) molecules. For normal B-cell development, the epigenetic programming is highly controlled and any profound perturbation of epigenetic mechanisms is linked to the pathogenesis of B-cell malignancy [25,26].

## 3. MiRNAs and Their Contribution to Tumor Microenvironment

MicroRNAs (miRNAs), a type of ncRNA, are considered one of the most potent epigenetic modulators [27,28]. MiRNAs function as major posttranscriptional gene regulators with critical roles in different cellular processes, such as differentiation, development, growth, angiogenesis, and apoptosis [29,30,31,32,33]. The ability of each miRNA to regulate several target genes simultaneously (more than 100 target genes) allows them to control multiple signaling pathways [34,35]. In past years, a special emphasis has been dedicated to an intense investigation on the function of miRNAs in cancer biology. Several studies have revealed the multifaceted roles of miRNAs in the different cellular pathways controlling cancer progression. Therefore, the aberrant levels of miRNAs during cancer initiation and progression could eventually sustain and enhance the different hallmarks of cancer [36,37,38,39].

The modulation of gene expression by miRNAs affects tumor progression either directly by regulating the phenotype of tumor cells or indirectly by modulating the TME. Particularly, miRNAs produced by either tumor or stromal cells can modulate TME via non-cell-autonomous mechanisms by influencing adjacent cells [8,40] with a profound impact on tumor progression [41]. Most of the studies on miRNAs in B-cell malignancies focused on the aberrant levels of miRNAs and their involvement in the pathogenesis of different lymphomas [42]. The miRNAs that have drawn early attention for their role in lymphomagenesis are miR-15a and miR-16-1 present at the 13q14 locus and reported to be frequently deleted in CLL patients [43,44]. This deletion accelerates the proliferation of B-cells. Other miRNAs that are found to be linked to CLL pathogenesis along with other types of B-cell malignancies are miR-150, miR-155, and miR-17-92 cluster that target and regulate the expression of vital transcription factors essential for the development of normal or malignant B-cells [45,46,47,48]. Moreover, the aberrant expression of hematopoiesis-specific miRNAs could impair the maturation and development of B-cells. An example of this effect is represented by miR-150 that is lymphopoietic-specific miRNA. Overexpression of this miRNA in hematopoietic/progenitor stem cells dramatically reduces the level of mature B-cells by blocking the transition from pro-B to pre-B cell stage and thus impairing B-lymphopoiesis [49]. Although the exact molecular pathway of this inhibitory effect was not delineated, one of the potential targets listed for miR-150 is the transcription factor FOXP1 that is another important transcriptional regulator of B-cell development [50]. The aberrant levels of miRNAs in B-cell malignancies and their contribution to the pathogenesis have been extensively studied with particular interest on their contributions to the different fundamental pathways of lymphomagenesis. 

Major changes in the miRNAs expression profile (miRNome) occur in the microenvironment that surrounds malignant B-cells. In the study by Willimott and Wagner, 2012, a miRNA array was applied to compare miRNome expression in CLL cells isolated from peripheral blood and CLL cells cultured on stromal cell layer [51]. The results revealed a profound change in the miRNome expression, where 20 miRNAs were significantly upregulated and induced by the stromal cell culture. Interestingly, this stromal effect was not only detected for individual miRNAs but also for clusters, where miR-125b, miR-99a, and let-7c encoded by the same cluster 21q21 were significantly up-regulated by the stromal effect. Another comparison in this study was performed to confirm the enhancing effect of the different stromal components on the expression of miRNAs, where miRNAs expression of CLL cells cultured on stromal cell layer supplemented with CD154 was compared with those cultured on stromal cell layer only. It was found that CD154 was able to enhance the expression levels of miRNAs of the same cluster (miR-17-92 cluster) and miRNAs present on the same locus (miR-212 and miR-132) to higher levels than cells cultured without CD145, revealing that the components of the stroma could differently modulate the expression of miRNAs. These results suggest that CLL cells in tissues (bone marrow or lymph nodes) surrounded by stroma show a different pattern of miRNAs expression than CLL cells circulating in the blood, suggesting that the stroma effect on miRNAs expression should be considered in therapeutic strategies. The mutual interaction between tumor cells and their microenvironment is one of the fundamental pathways that could be mediated and regulated by miRNAs.

In the present review, we are providing a comprehensive overview of the interplay between miRNAs and TME components in the different B-cell lymphomas and the involvement of this interplay in diagnosis, disease progression, and its reflection in treatment resistance. 

## 4. Involvement of TME-Associated miRNAs in Diagnosis and/or Prognosis 

In the endeavor to identify diagnostic and prognostic miRNAs that have significance in B-cell lymphomas and at the same time related to stroma, several global miRNAs profiling on multiple types of B-cell lymphomas were conducted. In the study by Iqbal et al., 2012 aiming to identify a reliable miRNAs classifier signature for MCL, a wide-scale miRNA profiling analysis of different types of B-cell lymphoma, normal B-cell subsets, and stromal cells were performed [52]. The miRNA profiling of the different B-cell lymphomas revealed two prominent miRNA signatures. One signature is highly distinctive of DLBCL and BL with a list of miRNAs referred to as stromal miRNAs (13 miRNAs). The other signature is referred to the miRNAs of the naive-and resting B-cells (6 miRNAs) and is more representative of MCL and small lymphocytic lymphoma (SLL) patients. Interestingly, these two signatures reflect the different features of TME in B-cell lymphomas, where MCL and SLL are characterized by low stromal content and low proliferation and aggressiveness compared with DLBCL and BL. To set a more precise miRNAs classifier for MCL subtypes and correlate it with the gene-expression profiling and clinical outcomes, an unsupervised hierarchical clustering of the global miRNAs profiling of only MCL patients were performed and three clusters were identified. Cluster (A) patients with high proliferation gene signature (PS) are characterized by an upregulation in proliferation regulator miRNAs (miR-17-92 cluster and its paralogs, miR-106a-363, and miR-106b-25). Patients of cluster B and C, with medium and low PS, respectively, showed significant upregulation in the stromal-associated miRNAs: miR-636, miR-539, and miR-485-3p for cluster B and miR-23a, miR-23b, let-7c, let-7-b, and miR-125b for cluster C patients. This suggests the involvement of stromal miRNAs in producing inhibitory proliferative signals in MCL patients. Interestingly, both clusters B and C have high expression of transcripts encoding proteins of the extracellular matrix. These results suggest that stromal-associated miRNAs could help in the diagnosis and subtyping classification of MCL [52]. The same research group also identified miR-542-3p as a stroma-related miRNAs and a classifier for robust classification of DLBCL subgroups [53].

The TME of FL is characterized by a significant number of immune cells (T-cells), corresponding to almost 50% of the total number of TME cells [21]. Therefore, in the attempt to identify miRNA signature for FL patients, the miRNA expression profile of FL patients was compared with the immune cells of the TME and CD4+ and CD8+ T-cells of healthy donors. Heat-map analysis showed a differential expression of miRNAs divided into three clusters. One of these clusters included miRNAs expressed at a high level in both FL and T-cells. This cluster was found to have higher expression of the T-cell characteristic markers CD3 and CD28, revealing that the main contributor of this list of miRNAs is the immune cells. The expression levels of two members of this list, miR-342 and miR-370, were strongly correlated with the levels of CD3 and CD28, suggesting that these miRNAs could represent a signature for FL enriched with T-cells in the microenvironment [54]. Previous studies reported the association between T-cells and miR-342 or miR-370. For example, miR-342 was previously found to be highly expressed in CD4+ T-cells than B-cells [55,56] connecting the expression of this miRNA to T-cell subsets and revealing a possible influence of immune T-cells on the miRNAs expression profile of FL patients.

Much of the research on miRNAs as biomarkers has focused on their use as diagnostic signatures and to a lesser extent as disease response biomarkers. One of the concerns is the potential non-specific elevation of miRNAs resulting from tumor cell lysis following treatment. However, miRNAs originating from non-malignant tumor-infiltrating cells could also reflect the disease response [57]. MiRNAs that are implicated in regulating the activity of the immunosuppressive components of TME, such as monocytic myeloid-derived suppressor cells (moMDSCs) and tumor-associated macrophages (TAMs), are considered disease response biomarkers in patients with DLBCL [58]. MDSCs are a diverse group of non-lymphoid immune suppressor cells deriving from the myeloid lineage and they function by orchestrating the TME and suppressing anti-tumor immune responses. MDSCs are divided into two groups of cells, monocytic (M-MDSCs) and granulocyte (G-MDSCs), which have strong immunosuppressive functions. Studies on these immune suppressor cells showed that several miRNAs, such as miR-494, miR-21, miR-30a, miR-155, miR-17-5p, miR-20a, miR-690, and miR-101, could regulate the proliferation, differentiation, and activation of these myeloid cells [59,60,61,62]. In the study by Cui et al., the elevated plasma levels of the MDSCs-associated-miRNAs, miR-494 and miR-21, in DLBCL patients profoundly decreased following 3-6 months of conventional first-line immuno-chemotherapy [58]. Moreover, the kinetic of the reduction of miR-494 expression level was associated with the interim-PET/CT status of the patients. The putative targets of miR-494 suggested that this miRNA could be a master regulator of the transcriptional modification pathways (i.e., chromatin remodeling (ASXL2, YY1) and DNA repair (BRIP1)) delineating the moMDSC immunosuppressive activity. Worth to note, both miR-494 and miR-21 have been previously referred as disease response biomarkers in CHL patients by the same research group, where the plasma levels of miR-494 and miR-21 in addition to miR-1973, reflected the kinetic of therapy response, where the significant reduction of their levels was more prominent in patients achieving complete versus partial treatment response. These miRNAs were referred to as node-associated miRNAs and they were among the top differential hits in a microarray profiling of CHL samples and nonmalignant lymph node samples [57]. Further studies are still needed to explore whether miRNAs originating from or associated with TME could represent efficient disease response biomarkers in different types of B-cell malignancy due to their specificity and sensitivity.

## 5. Involvement of TME-Associated miRNAs in Proliferation and Survival of Tumor Cells

Development, proliferation, and survival of B-cell malignancy are not only driven by genetic changes, but also by the crosstalk with the microenvironment components. The interplay machinery between malignant B-cells and stromal cells or non-neoplastic immune cells is involved in regulating B-Lymphomagenesis and significantly associated with survival and progression [63,64]. Several studies have been performed to dissect the molecular interactions between malignant B-cells and stromal cells (accessory cells). Most of these studies evaluated cell-to-cell interaction in co-culture assays using various bone marrow stromal cells (BMSC) to mimic the malignant cell crosstalk within the TME. Most of these studies found a remarkable change in gene expression profile of the malignant cells, pointing out the vital role of stromal cells in modulating key survival genes [65,66,67,68,69].

FDCs are stromal cells found in both primary and secondary lymph follicles of the B-cell areas of the lymphoid tissue and modulate the migration and differentiation of B-cells through several survival factors, cytokines, chemokines, and adhesion molecules. The cell-to-cell contact between B-cells and FDCs is a critical step for the maturation and survival of both normal and malignant B-cells [70]. In the study by Lin et al., co-culturing B-cells with FDC allowing cell-adhesion significantly modulated the expression of the master regulators of terminal B-cell differentiation, and this effect was mediated by an aberrant expression of specific miRNAs [71]. The interaction between FDC-like cell line (HK cells) and the DLBCL cell line (SU-4 lymphoma cells) significantly elevated the levels of miR-30a, b, c, and d and reduced those of let-7a and miR-9 levels. These miRNAs have direct targeting sites and regulation effects on the transcription factors B-cell lymphoma 6 (BCL6) and PR domain containing 1 (PRDM1), respectively. BCL-6 and PRDM1 are differentiation-related transcription factors that regulate B-cell differentiation and modulate lymphomagenesis [72]. The interplay between these transcription factors and stromal cells, mediated by miRNAs regulation, orchestrates B-cell maturation and lymphomagenesis [71].

Clonal expansion, survival, and proliferation of CLL occur in niches of the lymphoid tissues and the bone marrow. This supportive microenvironment provides contact-dependent stimuli with stromal endothelial cells, nurse-like cells and activated CD4+ T cells expressing CD40 ligand, in addition to an enhanced expression of several chemokines and cytokines that have been reported to regulate survival and proliferation of CLL cells [73,74,75]. This contact-dependent survival effect is partially mediated by miRNAs regulating activity. In the following section, we will list some findings that support the miRNA modulation effect on cell survival through tumor cell adhesion with the surrounding niche. 

Co-culturing CLL cells from patients’ peripheral blood with BMSC showed a prominent effect on CLL viability accompanied by a profound change in gene expression profile. Among the significantly upregulated genes, there is the lymphoid proto-oncogene TCL1 that functions as a co-activator of several pro-survival genes. The suggested mechanism that could explain the robust increased TCL1 expression was related to the significant decrease in the expression of the negative modulators of TCL-1 miR-29b, miR-181b, and miR-34b that directly target the 3’ UTR of TCL-1, indicating that their epigenetic regulatory effect could be involved in the genetic changes following CLL-stroma interaction [69]. 

T-cells or accessory cells within lymphoid TME express cytokines of the TNF ligand superfamily, such as B-cell–activating factor (BAFF), proliferation-inducing ligand (APRIL), or CD40-ligand (CD154), which are responsible for providing survival signals for B-cells. These survival signals originated from the microenvironment were found to directly contribute to the upregulation of miR-155 [76]. In the study by Cui et al., CLL cells that were stimulated with BAFF or CD154 expressed higher levels of miR-155 [76]. Interestingly, a higher level of miR-155 was found to be associated with an aggressive CLL phenotype and adverse clinical outcomes in patients. This association was mostly associated with the reduction of the direct target of miR-155, the Src homology-2 domain containing inositol 5-phosphatase 1 (SHIP1) protein. This phosphatase is encoded by INPP5D and inhibits BCR signaling [77], revealing that elevated level of miR-155 from CLL interaction with surrounding microenvironment could physiologically activate the BCR signaling and functionally modulate the proliferation and survival of malignant B-cells [76]. MiR-155 is one of the most prominent miRNAs that are repeatedly reported to be involved in innate immune responses and also linked to B and T-cell immunity [78,79,80].

Interleukin-21 (IL–21) was reported to manage the crosstalk between CLL cells and the surrounding microenvironment by modulating the expression of genes and miRNAs that control cell survival and proliferation. CLL cells that were pre-activated by the co-culture with activated CD4+ T cells expressing CD40 ligand to mimic TME and then stimulated by IL-21, showed a profound alteration in the expression of several chemokines involved in the dynamic interaction between tumor cells and TME. Particularly, Th2-related pro-inflammatory chemokines (CCL3, CCL4, CCL3L1, CCL22, and CCL17) were significantly reduced and Th1-related chemokines (CXCL9 and CXCL10) were significantly elevated, in addition to the reduction of other functional genes that are known to be involved in signaling pathways controlling survival and proliferation, such as CD40, DDR1, and PIK3CD [81]. This effect was mediated by modulating the expression of 63 miRNAs. Integrative analysis revealed that nine of these miRNAs could regulate almost 73% of the expressed genes. Among these miRNAs, miR-663b specifically targets and regulates a set of genes that were reported to be modulated by IL-21, such as CCL17, CD40, DDR1, and PIK3CD. These data reveal that IL–21 can regulate the expression of genes responsible for CLL survival and proliferation through mechanisms involving the modulation of specific miRNAs. This evidence is another proof that the modulation of genes regulating the crosstalk between CLL cells and TME occurs through mechanisms involving regulation of miRNAs [81,82].

Another miRNA with an impact on the regulation of the intercellular crosstalk between B-cell lymphoma and TME is miR-30a, which mediates the MDSCs-lymphoma progression effect [62]. The expression of miR-30a is significantly elevated in splenic MDSCs (monocytic and granulocytic) in mice with B-cell lymphoma. Controlling the expression of miR-30a (ectopic overexpression or inhibition) was able to regulate the differentiation of MDSCs, modulate their immunosuppressive function, and eventually control B-cell lymphoma progression. The administration of miR-30a agomir in a B-cell lymphoma mouse model was able to enhance tumor development, whereas miR-30a antagomir reduced tumor development [62]. The Suppressor of Cytokine Signalling-3 (SOCS3) is a direct target of miR-30a and was previously reported as one of the negative regulators of MDSCs differentiation through controlling and suppressing the JAK-STAT signaling pathway [83]. Therefore, miR-30a was considered as a critical mediator between one of the TME supportive components (MDSCs) and B-cell lymphomagenesis.

Epstein–Barr virus (EBV)-miRNAs are also involved in regulating the surrounding microenvironment of the infected B-malignant cells. EBV is a member of the human gamma-herpesvirus family that is involved in human malignancies, including several types of lymphomas [84,85]. MiRNAs encoded by EBV are believed to be involved in the EBV-induced lymphomagenesis. EBV encodes around 44 miRNAs that belong to two different classes: BART and BHRF1. These miRNAs contribute to the pathogenic effects of EBV through directly targeting cellular and/or viral mRNAs and thus interfering with several important cellular mechanisms, such as survival, proliferation, immune escape, and apoptosis. Moreover, EBV-miRNAs can regulate the surrounding microenvironment of the infected lymphoma cells with strong implications in cancer immunosurveillance [84,85,86,87]. For instance, in EBV-associated NHLs, the increased levels of BHRF1-3 were inversely correlated with the expression of its direct cellular chemokine target CXCL-11/I-TAC [88]. CXCL-11 is one of IFN-γ-inducible T-cell chemokines that mediates the immune cell trafficking into TME by acting as a potent T-cell chemo-attractant. The interaction between this chemokine with its receptor (CXCR3) could recruit immune cell subsets into the TME [89]. In the study by Xia et al., the suppression of this chemokine by the EBV-encoded miRNA (BHRF1-3) may function as an immunomodulatory mechanism in primary EBV+ lymphomas. Alternatively, the therapeutic targeting of BHRF1-3 represented an immunotherapeutic tool for enhancing the immunosurveillance of EBV-related tumors by controlling the cytotoxic T-cell cytokine networks [88].

## 6. Involvement of TME-Associated miRNAs in Cancer Therapy Resistance

One of the major clinical challenges of B-cell malignancy is the development of resistance to different treatments that leads to a higher probability of transformation into a more aggressive tumor with a greater ability to metastasize to other organs. Recent studies revealed that the aberrant expression of particular miRNAs has been linked to the development of treatment resistance mechanisms in B-cell malignancy. Stroma–lymphoma interactions not only influence lymphoma cell development and proliferation but also control and confer their resistance to chemotherapy drugs [90]. The importance of bone marrow microenvironment in regulating lymphoma apoptosis induced by drugs is extensively reported in vitro and in vivo [66,91,92,93]. These studies elucidated the reason why bone marrow is considered the preferred site for tumor relapse following conventional treatments. Experiments involving co-culture of lymphoma and stromal cells revealed that cell-to-cell interaction contributes to the acquisition of a multidrug-resistant phenotype [94]. Therefore, disruption of tumor-stroma interaction may enhance the efficacy of cytotoxic agents [94,95]. Several molecular targets control the stroma-mediated drug resistance acquisition and among them, miRNAs play a vital role.

An example of the cell adhesion-mediated drug resistance acquisition is the significant reduction in the pro-apoptotic protein BIM11 that is measured upon the interaction of MCL cells (Jeko-1 and Mino) or germinal center DLBCL cells (SUDHL-4) with stromal FDCs-derived cells (HK) [96]. This effect was accompanied by a miRNA regulatory activity. Particularly, lymphoma-stromal FDC interaction induces the expression of miR-181a, a direct post-transcription regulator of the BIM11 [96]. Interestingly, the reduction of the levels of miR-181a increased cell lymphoma apoptosis and sensitized their response to the cytotoxic drug mitoxantrone. Another set of miRNAs that are involved in the cell-to-cell interaction between stromal FDCs and B-cell lymphoma are miR-548 family members (miR-548f, miR-548h, and miR-548m). They are significantly downregulated following the co-culture of MCL cells (Jeko-1) or DLBCL cells (SUDHL-4) with stromal FDCs cells (HK), and miR-548m was reported as the most downregulated among the miR-548 members. MiR-548m was functionally associated with the stroma-mediated drug resistance acquisition. Indeed, bone marrow and lymph node stromal cells promoted MCL and other NHL growth, survival, and drug resistance via miR-548m/HDAC6 pathway. The cell-to-cell interaction between lymphoma and stromal cells induced HDAC6 expression as a result of the significant reduction of miR-548m, confirming that HDAC6 is a direct target for miR-548m. Furthermore, the stroma-lymphoma cell-to-cell interaction triggered a feed-forward loop of c-Myc/miR-548m, where activated c-Myc, recruited EZH2 complex (a co-repressor of miR-548m promoter) inducing a sustained reduction of miR-548m and a consequent upregulation of HDAC6. Both ectopic expression of miR-548m and knockdown of HDAC6 were able to disrupt the stroma-lymphoma adhesion and consequently induce cell apoptosis, suppress colony formation, sensitize B-lymphoma cells to the cytotoxic drug mitoxantrone, and abolishing the cell adhesion–mediated drug resistance acquisition (CAM-DR) [97]. Other TME-associated miRNAs that have been reported to be involved in treatment resistance are miR-221 and miR-222, whose expressions were decreased following the co-culturing of acute lymphoblastic leukemia (ALL) cells with human bone marrow cells including BMSC and primary human osteoblasts. Reduction of miR-221 and miR-222 levels was accompanied with an elevation in the expression of their target gene p27 (CDKN1B). Leukemic cells with high expression of p27 are more resistant to Cytarabine and Vincristine chemotherapies. On the other hand, the ectopic expression of miR-221 during the co-culturing of ALL cells with the human bone marrow cells significantly sensitized the cells to these cytotoxic drugs [98].

One of the most extensively studied miRNA is miR-155, which is involved in many cellular processes including drug resistance. Recently, we demonstrated the role of miR-155 in regulating chemoresistance in lung cancer, CLL, and ALL [78,99]. Overexpression of miR-155 leads to enhanced cell viability and growth in cisplatin-treated immunoblastic B-cell leukemia/lymphoma cells (JM1) and doxorubicin treated ALL cells [78,99]. Moreover, we showed that TP53 is involved in a negative feedback loop with miR-155 and showed that this feedback mechanism is involved in resistance to multiple types of chemotherapeutic agents in various tumors [78,99]. In contrast, some studies have demonstrated that miR-155 can modulate the chemo-sensitivity of DLBC cells, where the expression of miR-155 was associated with vincristine resistance. Vincristine-resistant DLBC cell lines have a low level of miR-155, whereas vincristine-sensitive DLBC cells have a high level of miR-155 [100].

In another study, after co-culturing lymphoma cells with immune cells and CD8+T cells, miR-155 inhibition increased the percentage of CD8+T cells in human B-lymphoma [101]. In contrast, overexpression of miR-155 in human B-lymphoma cells decreased the percentage of CD8+Tcells, as well as enhanced CD8+Tcell apoptosis [101]. These molecular mechanisms of the interactive regulation between tumor and stroma components mediated by miRNAs, could offer an explanation of the detected resistance of lymphoma cells toward the widely used drugs for lymphoma therapy and also offer potential novel therapeutic targets.

## 7. Crosstalk of Exosomal MiRNAs and TME in B-Cell Malignancy 

Cells can release a variety of small membrane-derived vesicles (EVs) that include ectosomes, microparticles, microvesicles (MVs), tumor-derived MVs (TMVs), and exosomes [102,103,104,105,106]. Exosomes (30-140 nm in size) are the best-characterized class of EVs. They are secreted by almost all cell types including mast cells, dendritic cells, B lymphocytes, neurons, adipocytes, endothelial cells, and epithelial cells [78,103,107,108,109]. It has been shown that exosomes can carry different types of cargos from their donor cells, including proteins and nucleic acids (DNA, mRNAs, and non-coding RNAs) involved in cell-to-cell communication. This crosstalk in the TME can regulate tumor growth, metastasis, epithelial-mesenchymal transition (EMT), angiogenesis, and immune functions through the modulation of specific target genes. Recent studies have revealed that cancer cells release a high level of exosomes comparing to normal cells and can transfer functional information at the paracrine level by shuttling exosomes loaded with biological active molecules [78,103,107,108,109]. Although the exact role of exosomal miRNAs in hematologic malignancies is not extensively addressed, some studies suggest a major contribution of exosomes (especially CLL-derived exosomes) in promoting tumor progression through modulating stromal cells [110,111,112]. CLL is the most studied B-cell malignancy regarding exosomes involvement in pathogenesis. CLL cells were found to release more exosomes in plasma comparing to normal B-cells. Moreover, CLL derived exosomes are enriched in unique miRNA content compared with normal B-cells. For instance, CLL-derived exosomes extracted from plasma of CLL patients showed significant overexpression of miR-150, miR-155, and miR-29 compared with exosomes from healthy donors. Interestingly, these exosomes are regulated by the B-cell receptor (BCR) signaling pathway that play an important pathogenic role in CLL and by TME [112]. CLL-derived exosomes enriched with miR-150 and miR-146a were able to remodel the TME and enhance stromal cell proliferation and migration and eventually promote CLL survival and growth. This effect was elucidated in the study by Paggetti et al., where CLL-derived exosomes were able to actively deliver their content (miRNAs and proteins) to benign stromal cells in the TME, such as mesenchymal stem cells (MSCs) and endothelial cells, inducing an inflammatory phenotype of these cells and acquisition of a cancer-associated fibroblasts (CAFs) phenotype [111]. Consequently, these stromal cells exhibited higher rates of proliferation and migration that was accompanied by an elevated secretion of inflammatory cytokines and chemokines (IL-8, IL-6, IL-32, IL-34, CXCL1, CCL2, CCL5) that efficiently contributed to a tumor-promoting microenvironment.

Another example of exosome-mediated communication in the TME is represented by CLL-derived exosomes that are actively taken up by normal stromal cells (HS-5). These exosomes were found to be enriched with a high level of miR-202-3p and their transfer induced elevated levels of this miRNA in the recipient stromal cells (HS-5), resulting in a significant reduction of its target gene Suppressor of Fused (Sufu). This interesting finding suggests that CLL cells may specifically and actively load miR-202-3p into exosomes to influence the expression of Sufu in recipient cells [110]. Sufu is overexpressed in patients with poor prognosis and reported to be involved in CLL pathogenesis by regulating Sonic Hedgehog Signaling Pathway [113,114].

Additionally, in prior study, it was found that 19 miRNAs were dysregulated in human MSCs after treatment with MM cell-conditioned media. High expression levels of miR-146a and other dysregulated miRNAs resulted in an altered profile of chemokines and cytokines, including elevated levels of the chemokines C-X-C motif ligand 1 (CXCL1), C-C motif ligand 5 (CCL5), monocyte chemoattractant protein-1 (MCP-1), interleukin 8 (IL-8), interleukin 6 (IL-6), and interferon gamma-induced protein 10 (IP-10). MM-derived exosomes that contained miR-146a were further shown to be transferred to MSCs and drive the aforementioned changes [115,116].

In a similar study, it was found that miR-146a and miR-21 were enriched in exosomes derived from the MM cell line OPM2. MM-derived exosomes contained miR-146a and miR-21 could increase cell proliferation, IL-6 production, and cancer-associated fibroblast (CAF) transformation of MSCs after co-culture of MSCs with OPM2 conditioned media [116,117]. Similarly, in another study, multiple myeloma BM mesenchymal stromal cell (MM-BMSC)-derived exosomes were shown to play a role in MM disease progression, in vivo. It was shown that exosomes isolated from BM-MSCs of patients with MM promoted MM tumor growth, in vivo. Moreover, exosomal miR-15a level was significantly increased in normal versus MM BM-MSC–derived exosomes and miR-15a–containing exosomes inhibited MM cell proliferation [118,119].

Collectively, these findings demonstrate a widespread role of exosome-mediated interactions between the TME and B-cell malignancies (Figure 2). Therefore, exosomes constitute a novel mechanism for intercellular transfer of genetic materials in hematological malignancies. However, more studies are still to be conducted to explore and identify the exosomal miRNAs signature in B-cell malignancies and uncover their function.

## 8. Conclusions and Future Perspective

Recent progress in the miRNA research field has identified many aspects of miRNAs in cancer that can act either as oncogenes or as tumor suppressor genes or sometimes as both. Several reports have highlighted the significance of miRNAs in regulating the TME generally and tumor stromal cells specifically. The functional roles of miRNAs in tumor microenvironments have been well studied in solid tumors; however, we still need to have a clear overview of the miRNA-mediated regulation of TME in B-cell malignancies.

An increasing number of miRNA studies were found to have the potential application as diagnostic or prognostic biomarkers for B-cell malignancies and currently, several clinical studies on the therapeutic use of miRNAs are ongoing. An example is the clinical trial (Identifier: NCT03000335) with ALL patients which evaluates the differential levels of specific miRNAs to predict ALL relapse risk.

In addition, miRNA inhibitor-based therapies are currently in clinical trials. An example is MRG-106 therapy that is an oligonucleotide inhibitor of miR-155, which is being tested in phase 1 clinical studies in different cancer types including Cutaneous T-cell Lymphoma, CLL, DLBCL and Adult T-Cell Leukemia/Lymphoma (clinicaltrials.gov identifier NCT02580552).

Overall, the future direction in the treatment of patients with B-cell malignancies could be the combination of conventional therapies along with miRNA-based therapeutics to synergistically block several pathways and eventually reduce the severity of side effects. Further preclinical and clinical studies with miRNA-based therapeutics are needed to provide various possibilities for targeted therapy.

Moreover, a complete understanding of the role of miRNAs in the TME of B-cell malignancies could also help the development of efficient therapeutic strategies, as therapies modulating the microenvironment surrounding tumor cells were found to be able to slow tumor progression and prolong patients’ survival, especially when used in combination with chemotherapies.

In conclusion, in the present review, we focused on elucidating the functional involvement of miRNAs in the interplay between tumor cells and their surrounding microenvironment. This involvement was either evaluated in a diagnostic and/or prognostic perspective or used to control the proliferation and survival of tumor cells or even associated with tumor resistance (Table 1). We aimed to retrieve from the different research studies the prominent miRNAs that have been associated with the TME in B-cell malignancies and presenting them as innovative targets for clinical application.

## Figures and Tables

**Figure 1 ijms-21-03387-f001:**
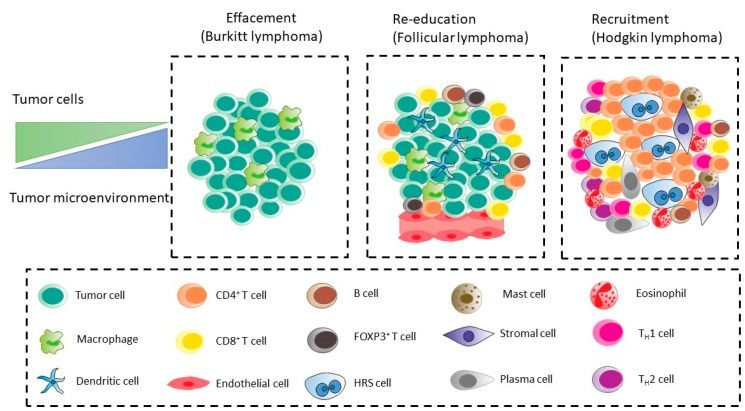
Schematic overview of the “recruitment,” “re-education,” and “effacement” tumor microenvironment models of the three B-cell lymphoma subtypes. The tumor microenvironment (TME) of B-cell lymphoma subtypes represents variable tumor cell content (from ~1% in classical Hodgkin lymphoma (CHL) to 90% in Burkitt lymphoma (BL)). (Abbreviations: HRS: Hodgkin Reed–Sternberg, FOXP3: Forkhead box protein P3, TH: T helper).

**Figure 2 ijms-21-03387-f002:**
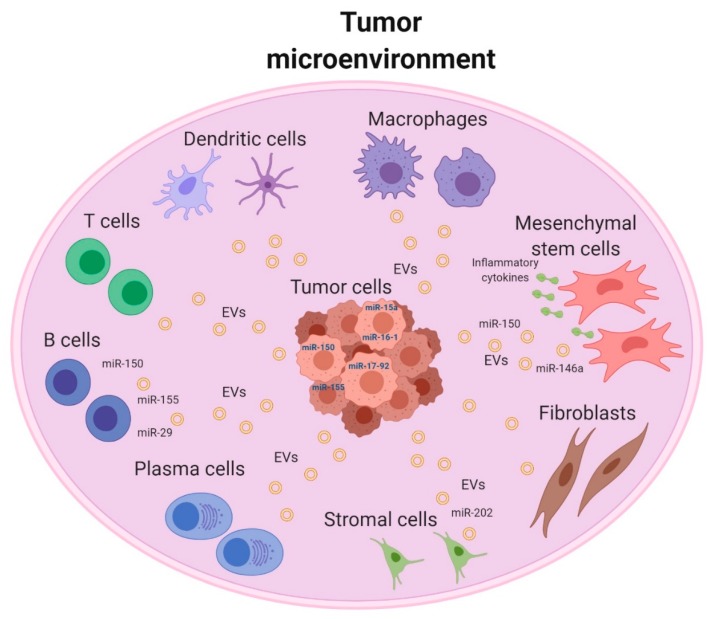
Schematic overview of tumor microenvironment of B- cell malignancies. Tumor cell-derived exosomes contain specific microRNAs (miRNAs) that can participate in the generation of a tumor microenvironment through paracrine signaling. (Figure is created with BioRender.com).

**Table 1 ijms-21-03387-t001:** List of miRNAs that are involved in the interplay between TME/stroma and tumor cells in the different B-cell malignancies.

	Type of B-Cell Malignancy	TME/Stoma Associated MiRNAs	Function	Ref
**Diagnostic and prognostic function**	Mantle cell lymphoma	miR-23a, miR-23b, let-7c, let-7-b, miR-125b, miR-636, miR-539, miR-485-3p	Stroma-associated miRNAs is correlated with proliferation gene signature for MCL subtyping.	[52]
	Diffuse large B-cell lymphoma	miR-542-3p	Stromal miR-542-3p was set for the robust classification of DLBCL subgroups.	[53]
	Follicular lymphoma	miR-342 and miR-370	These miRNAs act as signature marker for FL enriched with CD4^+^ T-cells in the microenvironment.	[54]
	Diffuse large B-cell lymphoma	miR-494 and miR-21	The MDSCs associated miRNAs were set as disease response biomarkers in patients with DLBCL.	[58]
**Controlling diseases proliferation and survival**	Chronic lymphocytic leukemia	miR-29b, miR-181b, miR-34b	The decrease expression of these TCL1-regulatory miRNA’s following co-culturing of CLL cells with stromal cells is partially responsible for enhancing pro-survival signaling molecules.	[69]
	Chronic lymphocytic leukemia	miR-155	Upregulation of miR-155 by accessory cells of lymphoid tissue microenvironment is associated with activated BCR signaling and a more aggressive disease.	[76]
	Diffuse large B-cell lymphoma	miR-9, let-7, miR-30a, b, c and d	The elevated expression of these miRNAs following the adhesion of B-lymphocytes and FDCs regulates B-cell survival and differentiation by targeting the regulators of terminal B-cell differentiation PRDM1 and BCL6.	[71]
	B-cell lymphoma mouse model	miR-30a	Regulate the differentiation of MDSCs and modulate their immunosuppressive function and eventually control B-cell lymphoma progression	[62]
**Mediating drug resistance**	EBV-associated non–Hodgkin’s lymphomas (Diffuse large B-cell lymphoma and Burkitt lymphoma)	EBV-miRNA BHRF1-3	The targeting suppression of ebv-miR-BHRF1-3 to its putative target gene, T-cell attracting chemokine CXCL-11/I-TAC, functions as an immunomodulatory mechanism in EBV-related lymphomas.	[88]
	Mantle cell lymphoma and Diffuse large B-cell lymphoma	miR-181a	The elevation in miR-181a following adhesion of lymphoma cells to FDCs enhanced the drug resistance mechanism toward mitoxantrone through reducing the pro-apoptotic protein BIM11.	[96]
	Mantle cell lymphoma and Diffuse large B-cell lymphoma	miR-548m	The downregulation of miR-548m contributes to the stroma-mediated cell survival and mitoxantrone resistance through HDAC6 upregulation and a c-Myc/miR-548m feed-forward loop.	[97]

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
