# Peer review of "The Interplay between MicroRNAs and the Components of the Tumor Microenvironment in B-Cell Malignancies"

_ijms, 2020, doi:10.3390/ijms21093387_

Round 1

Reviewer 1 Report

The research paper focuses on microRNAs relationship with components of tumor microenvironment that affects B-cell malignancies.

Minor comments:

A reason for illustrating the differences between the tumor microenvironment of different cancers is warranted, similarly to the way different microenvironments within B-cell malignancies are highlighted as being important to consider.  (Lines 55-59, 79-85)

It is suggested that the introduction gives a brief synopsis on B cell malignancies. Authors introduce tumor microenvironment and miRNAs, but not B-cell cancers, a main focus of this paper. This will aid in creating a flow to the paper as opposed to directly talking about the relationship between B-cell cancers with TME and miRNAs.

In regards to section 3 (paragraph 2) a summary is given, “The mutual interaction between tumor cells and their microenvironment”, that better fits with paragraph 3 as paragraph 2 focuses on aberrant gene expression of miRNAs that modulates B-cell cancers.

What is the correlation of these miRNAs (miR-494, miR-21 etc.) as disease response biomarkers? (Lines 190-192)

At the beginning it is stated that IL-21 modulates miRNAs, it is unclear based on the way it is described if IL-21 modulates miR-663b or that they target the same genes. (Lines 260-273)

For table 1, it is suggested that the authors include the specific drugs that are targeted by miR-181a (typo “mi-181a”) and miR-548 as done in text to better make connections. Overall, this table should have clear boundaries as to what miRNAs are involved in the different functions as it is unclear at first glance.

Author Response

Point-by-point response to reviewers’ comments

First, we thank the reviewers for their constructive comments. We extensively revised the text and addressed all the points raised by the reviewers.

Reviewer #1

The research paper focuses on microRNAs relationship with components of tumor microenvironment that affects B-cell malignancies.

1- A reason for illustrating the differences between the tumor microenvironment of different cancers is warranted, similarly to the way different microenvironments within B-cell malignancies are highlighted as being important to consider.  (Lines 55-59, 79-85).

Response:   We edited this part in text based on the reviewer’s comment.

2- It is suggested that the introduction gives a brief synopsis on B cell malignancies. Authors introduce tumor microenvironment and miRNAs, but not B-cell cancers, a main focus of this paper. This will aid in creating a flow to the paper as opposed to directly talking about the relationship between B-cell cancers with TME and miRNAs.

Response: According to reviewer’s comment, we have added a brief synopsis in the introduction part on the different B-cell malignancies that were covered in our manuscript.

 3- In regards to section 3 (paragraph 2) a summary is given, “The mutual interaction between tumor cells and their microenvironment”, that better fits with paragraph 3 as paragraph 2 focuses on aberrant gene expression of miRNAs that modulates B-cell cancers.

Response: According to reviewer’s comment, now this section was moved to paragraph 3.

4- What is the correlation of these miRNAs (miR-494, miR-21 etc.) as disease response biomarkers? (Lines 190-192)

Response: We edited this part in text based on the reviewer’s comment.

5- At the beginning it is stated that IL-21 modulates miRNAs, it is unclear based on the way it is described if IL-21 modulates miR-663b or that they target the same genes. (Lines 260-273)

Response: According to reviewer’s comment, this part has been rewritten in a more clarified way to avoid any confusion.

6- For table 1, it is suggested that the authors include the specific drugs that are targeted by miR-181a (typo “mi-181a”) and miR-548 as done in text to better make connections. Overall, this table should have clear boundaries as to what miRNAs are involved in the different functions as it is unclear at first glance.

Response: As the reviewer suggested, the drugs that are functionally involved with miR-181a and miR-548, have been listed in the table.  Moreover, the table is also formatted for clear boundaries between the different studies. 

Reviewer 2 Report

In this review the authors summarize literature focused on studying microRNAs involved in B-cell lymphomas pathogenesis, specifically the ones implicated in the crosstalk with the tumor microenvironment, a field in expansion in these lymphomas. These microRNAs may be used as diagnostic/prognostic markers and, importantly, may have implications on the acquisition of resistance to treatment.

The work is well organized in structure. I have the following suggestions to improve the manuscript:

  • there are some typos/grammar mistakes to be amended (examples are
    • line 19: has been
    • line 21: involved
    • line 39: consist
    • line 91: emphasis
    • line 164: contribute
    • lines 167-171: rephrase the section, it is a bit confusing the way it is
    • line 241: functions
    • line 340: EMT is not abbreviated here, in the first time it was written
    • (etc)
  • for microRNAs, avoid stating "expression" - since they are non-coding RNAs, stick with higher or lower levels
  • I suggest to include a brief description in the beginning of the most relevant B-cell malignancies, regarding some clinical aspects and clinical challenges in the field, for the reader to relate to the topic clinically
  • section 2: the authors describe some patterns of interaction between tumor cells and TME. This would be a very nice illustration for the review, relating to the types of interactions and the several lymphomas
  • authors should emphasize the clinical application of the knowledge described in the review in the conclusions

Author Response

Point-by-point response to reviewers’ comments

First, we thank the reviewers for their constructive comments. We extensively revised the text and addressed all the points raised by the reviewers.

Reviewer 2

In this review the authors summarize literature focused on studying microRNAs involved in B-cell lymphomas pathogenesis, specifically the ones implicated in the crosstalk with the tumor microenvironment, a field in expansion in these lymphomas. These microRNAs may be used as diagnostic/prognostic markers and, importantly, may have implications on the acquisition of resistance to treatment. The work is well organized in structure. I have the following suggestions to improve the manuscript:

1- There are some typos/grammar mistakes to be amended (examples are

line 19: has been  

line 21: involved      

line 39: consist         

line 91: emphasis   

line 164: contribute   

lines 167-171: rephrase the section, it is a bit confusing the way it

line 241: functions 

line 340: EMT is not abbreviated here, in the first time it was written  (etc)

Response: We have reviewed the manuscript and corrected errors in grammar, punctuation and language usage. We have also rephrased some sentences that sound unnatural to improve the flow of the texts.

2- For microRNAs, avoid stating "expression" - since they are non-coding RNAs, stick with higher or lower levels.

Response: As the reviewer suggested we replaced the term expression to (high/low level) in most of the cases except the ones that we were addressing the expression process itself.

3- I suggest to include a brief description in the beginning of the most relevant B-cell malignancies, regarding some clinical aspects and clinical challenges in the field, for the reader to relate to the topic clinically.

Response: As the reviewer suggested, we added a brief synopsis in the introduction part on the different B-cell malignancies that were covered in our manuscript.

4- Section 2: the authors describe some patterns of interaction between tumor cells and TME. This would be a very nice illustration for the review, relating to the types of interactions and the several lymphomas

Response: According to reviewer’s comment, now we have prepared a new figure that describes some patterns of interaction between tumor cells and TME in B-cell lymphoma subtypes.

5- Authors should emphasize the clinical application of the knowledge described in the review in the conclusions

Response: We have added small section about the clinical application of miRNAs in review in the conclusions.

Round 2

Reviewer 2 Report

No further comments.